# Population Genomics of Domesticated *Cucurbita ficifolia* Reveals a Recent Bottleneck and Low Gene Flow with Wild Relatives

**DOI:** 10.3390/plants12233989

**Published:** 2023-11-27

**Authors:** Xitlali Aguirre-Dugua, Josué Barrera-Redondo, Jaime Gasca-Pineda, Alejandra Vázquez-Lobo, Andrea López-Camacho, Guillermo Sánchez-de la Vega, Gabriela Castellanos-Morales, Enrique Scheinvar, Erika Aguirre-Planter, Rafael Lira-Saade, Luis E. Eguiarte

**Affiliations:** 1Consejo Nacional de Humanidades, Ciencias y Tecnologías, Av. Insurgentes Sur 1582, Col. Crédito Constructor, Ciudad de México 03940, Mexico; 2Departamento de Ecología Evolutiva, Instituto de Ecología, Universidad Nacional Autónoma de México, Circuito Exterior s/n Anexo al Jardín Botánico, Ciudad de México 04510, Mexico; josue.barrera@tuebingen.mpg.de (J.B.-R.); jaimegasca@yahoo.com (J.G.-P.); gsanchezdv@gmail.com (G.S.-d.l.V.); escheinvar@gmail.com (E.S.); eaguirre@ecologia.unam.mx (E.A.-P.); 3Department of Algal Development and Evolution, Max Planck Institute for Biology, Max-Plank-Ring 5, 72076 Tübingen, Germany; 4Unidad de Biotecnología y Prototipos, Facultad de Estudios Superiores Iztacala, Universidad Nacional Autónoma de México, Av. De Los Barrios 1, Col. Los Reyes Iztacala, Tlalnepantla 54090, Mexico; 5Centro de Investigación en Biodiversidad y Conservación, Universidad Autónoma del Estado de Morelos, Av. Universidad 1001, Col. Chamilpa, Cuernavaca 62209, Mexico; alejandra.vazquez@uaem.mx (A.V.-L.); andrea.loppc@gmail.com (A.L.-C.); 6Departamento de Conservación de la Biodiversidad, El Colegio de la Frontera Sur Unidad Villahermosa, Carretera Villahermosa-Reforma km. 15.5, Ranchería El Guineo 2a Sección, Villahermosa 86280, Mexico; gcastellanos@ecosur.mx

**Keywords:** demographic modelling, plant domestication, population genomics, gene flow, wild relatives

## Abstract

*Cucurbita ficifolia* is a squash grown from Mexico to Bolivia. Its ancestor is unknown, but it has limited compatibility with wild xerophytic *Cucurbita* from Mexico’s highlands. We assembled the reference genome of *C. ficifolia* and assessed the genetic diversity and historical demography of the crop in Mexico with 2524 nuclear single nucleotide polymorphisms (SNPs). We also evaluated the gene flow between *C. ficifolia* and xerophytic taxa with 6292 nuclear and 440 plastome SNPs from 142 individuals sampled in 58 sites across their area of sympatry. Demographic modelling of *C. ficifolia* supports an eight-fold decrease in effective population size at about 2409 generations ago (95% CI = 464–12,393), whereas plastome SNPs support the expansion of maternal lineages ca. 1906–3635 years ago. Our results suggest a recent spread of *C. ficifolia* in Mexico, with high genetic diversity (*π* = 0.225, *F_ST_* = 0.074) and inbreeding (*F_IS_* = 0.233). Coalescent models suggest low rates of gene flow with *C. radicans* and *C. pedatifolia*, whereas ABBA-BABA tests did not detect significant gene flow with wild taxa. Despite the ecogeographic proximity of *C. ficifolia* and its relatives, this crop persists as a highly isolated lineage of puzzling origin.

## 1. Introduction

The evolutionary process of plant domestication has long been considered a case study for understanding the forces that guide population differentiation and speciation [1]. Artificial selection has been used as a proof of concept to illustrate the power of natural selection in generating adaptation and driving diversification [2,3,4]. Species under domestication have also proved useful for analysing additional evolutionary forces that shape the levels and patterns of genetic diversity in the speciation continuum, including mutation, such as from point mutations to genomic rearrangements and polyploidy [5,6], genetic drift caused by reductions in effective population sizes [7], and gene flow/introgression [8,9,10].

The role of gene flow deserves special attention because it acts as an intermediary between the remaining evolutionary forces: gene flow allows mutations to spread among populations, enabling local selection regimes to act upon them; it can also favour the appearance of novel genetic combinations with increased fitness and can counteract the effects of genetic drift, minimizing divergence at neutral loci among subpopulations [11,12].

In the context of plant domestication, gene flow can either erode or favour selective gains in the crop, especially during the first stage of wild vs. domesticated population differentiation. When alleles associated with undesirable phenotypes originating in wild populations are continuously introduced to populations under domestication, gene flow can prevent divergence; thus, the emergence of reproductive barriers has been seen as an important condition for full domestication [13]. On the other hand, gene flow can contribute to maintaining connectivity among populations under human management, facilitating the spread and fixation of domestication syndrome alleles [14]. Such connectivity can also counteract the effects of genetic bottlenecks that may occur due to strong artificial selection, clonal propagation, and/or changes in the reproductive system favouring inbreeding [12,15,16,17].

Once domesticated populations have differentiated from their wild progenitors, secondary contact between the crop and closely allied subspecies or species (i.e., the secondary gene pool [18]) can lead to the introgression of local alleles into the crop’s primary gene pool, enabling crops to adapt and diversify (e.g., [19,20,21]).

Domesticated crops from the Mesoamerican centre of agriculture, such as maize, beans, tomato, and squashes, are also prone to experience historical and recent gene flow with their wild progenitors [22,23,24,25]. This phenomenon is promoted by the close coexistence of wild, intermediate (i.e., semi-domesticated), and fully domesticated forms in traditional agricultural systems as was first noticed by Vavilov [26].

Squash crops of the Neotropical genus *Cucurbita* (Cucurbitaceae) include six independently domesticated taxa distributed from Mexico to Bolivia, with Mexico as the centre of diversity of the genus [27]. Among domesticated taxa, *Cucurbita ficifolia* Bouché is adapted to highland habitats at elevations above 1000 m above sea level (Figure 1) and is not closely related to any known wild *Cucurbita* species [28,29]. This squash is known in English as the figleaf gourd and as *chilacayote* in Mexico. The figleaf gourd differs greatly from the remaining members of the genus by its seed colour and morphology (black seeds, although white seeds are occasionally found, with a smooth and rounded margin), pubescent filaments, and dimpling fruit surface [30]. Fruit morphological diversity is slightly higher in Peru when compared to Mexico [31], but cultural significance is deeper in Mexico (pers. obs.). The available genetic data on this crop are limited to isozymes and RAPDs, which detected low diversity levels [32,33]. *Cucurbita ficifolia* is a diploid species (2n = 40) as all the other studied species in the genus [34]. Linguistic evidence and pollinator associations point to a Mesoamerican origin of the crop, whereas the earliest archaeobotanical records are found in Coastal Peru (Figure 1; [30,32,35]).

The closest wild relatives of *C. ficifolia* are grouped in the *Cucurbita* xerophytic clade, also known as the *foetidissima* group, which is native to the highlands of northern and central Mexico (Figure 2a; [27]). The xerophytic clade includes *C. foetidissima*, *C. pedatifolia*, *C.* x *scabridifolia* (a putative hybrid of *C. foetidissima* and *C. pedatifolia*) [39], and *C. radicans* [28,29,40]. *Cucurbita foetidissima* and *C. pedatifolia* have been reported as diploids [34]; the *Cucurbita* genus displays a well-conserved karyotype of 2n = 40 and genome macrosynteny across species [23]. All domesticated *Cucurbita* are mesophytic, characterized by an annual life cycle and fibrous roots, whereas xerophytics are perennials with tuberous storage roots.

*Cucurbita ficifolia* is not compatible with other mesophytic domesticated taxa but has some limited reproductive compatibility with *C. foetidissima* and *C. pedatifolia*, which therefore represent the secondary gene pool of the figleaf gourd [41,42].

In this study, we assembled a high-quality reference genome for the figleaf gourd *C. ficifolia* and generated a robust dataset of single nucleotide polymorphisms (SNPs) that includes both nuclear and plastome (i.e., chloroplast’s genome) variants. Nuclear and plastome SNPs were used to evaluate the diversity and demographic history of *C. ficifolia* and to assess the occurrence of historical gene flow between this crop and its xerophytic relatives *Cucurbita foetidissima*, *C. pedatifolia*, *C. radicans* and *C.* x *scabridifolia*.

## 2. Materials and Methods

### 2.1. Sampling

Samples were collected across the distribution range of the taxa in Mexico (Figure 2). Fruits of *C. ficifolia* were bought at traditional markets where we could assess their geographical origin, whereas fruits of wild taxa were collected in the field. Seeds were sown in a greenhouse to obtain total DNA from leaf tissue with a CTAB protocol [43]. For *C. foetidissima*, the sampling was focused on central Mexico because this is its area of coexistence with *C. ficifolia* (Figure 2; Table 1). For *C. pedatifolia*, three samples from the northern area were reassigned a posteriori as *C.* x *scabridifolia*, as they displayed contrasting leaf morphologies, and their genetic constitution indeed grouped them with *C.* x *scabridifolia*. Allopatric *C. cordata* from Baja California was included as an outgroup. Our final dataset was based on 142 individuals sampled in 58 sites; sample sizes per taxon are shown in Table 1.

### 2.2. Genome Sequencing and Assembly

Total DNA was obtained from leaves of a seed grown from a *C. ficifolia* fruit collected in Morelos (Mexico) and sequenced with 21.7 Gb of Illumina HiSeq4000 (500 and 1000 bp paired-end libraries) and 23.4 PacBio Sequel (20 kb size-selected library). Quality filters were applied, and adapters were removed before merging paired reads. The chloroplast and mitochondrial genomes were assembled with NOVOplasty [44], using available organellar genomes of the genus as seeds [45,46]. *Cucurbita ficifiolia*’s organellar genomes were then used to separate Illumina nuclear from organellar reads, and the former were assembled de novo using Platanus [47]. The resulting contigs were assembled into larger contigs using the PacBio reads and DBG2OLC [48], followed by additional polishing steps. Reference-guided scaffolding was performed using RaGOO [49] against the genome of *C. maxima* [50] using PacBio corrected reads to correct misassemblies. The chromosome numbers were assigned in correspondence to the genome of *C. moschata* [50]. Finally, we performed a BUSCO analysis [51] against the *embryophyte odb9* database (please refer to details in the electronic Appendix A).

### 2.3. GBS Procedure

Samples were submitted to genotyping-by-sequencing (GBS) using enzymes NspI and BfuCI/Sau3AI and fragment size selection at 200–300 base pairs. Illumina (1 × 100 SE) sequences were obtained using a 94-sample multiplex protocol on a NovaSeq S1 FlowCell (University of Minnesota Genomics Center, Minneapolis, MN, USA), generating ca. 4 M reads/sample.

Raw reads were quality-filtered using Trimmomatic v0.39 [52] to remove adapter sequences, leading and trailing bases with Phred quality < 25, and cutting the read when the average quality per base was below 20 with a 4-base sliding window. Only reads with a total length > 60 bp were kept. Eight *C. ficifolia* samples did not attain the minimum quality and had to be excluded from downstream analyses, keeping a final sample size of *n* = 28 for this taxon.

### 2.4. Reference-Guided Read Mapping

After excluding reads that mapped to organellar genomes with *ipyrad* v.0.9.31 [53], nuclear reads were mapped to *C. ficifolia* reference genome. The first dataset was built for *C. ficifolia* samples to evaluate the genetic diversity of the crop and reconstruct its historical demography in Mexico. A second dataset was built with the five taxa under study and the outgroup for evaluating the hybrid nature of *C.* x *scabridifolia* and detecting possible gene-flow events between the crop and its wild relatives.

Single nucleotide polymorphisms (SNPs) were called with the following parameters: minimum read depth 6X, maximum cluster depth 1 × 10^4^, maximum 2 alleles per locus, a minimum of 4 samples per locus, and the remaining parameters as default. Datasets were filtered with *vcftools* v0.1.16 [54] to exclude sites with a proportion of missing data > 30%, a between-sites distance under 250 bp (to keep one SNP per locus), and sites out of Hardy–Weinberg Equilibrium below a 0.0001 threshold. We suppressed adjacent SNPs with a squared correlation coefficient (r^2^) larger than 0.25 within 100 kbp sliding windows with a step size of 100 bp to reduce bias due to linkage disequilibrium (with *plink* v1.90 [55]). The final *C. ficifolia* nuclear dataset consisted of 2524 SNPs (mean read depth = 14.42x, sd = 17.64x). The five-taxa nuclear dataset consisted of 6292 SNPs (mean read depth = 15.8x, sd = 10.7x).

For plastome data, we mapped the reads of each sample to *C. foetidissima* plastome KT898810 [56], using *ipyrad* with the same parameters previously described, except the maximum number of alleles per locus, which was set to 1 to exclude potential paralogs [57]. The dataset was filtered to keep only sites with 20% maximum missing data, a maximum read depth of 800x, and excluding sites from the inverted repeats regions. Our final plastome dataset consisted of 440 SNPs (mean read depth = 78x, sd = 46x), which were concatenated to obtain a single sequence per individual.

### 2.5. Data Analysis

*Figleaf gourd diversity and historical demography*—Genetic diversity and structure (*F* statistics [58]) estimates were calculated from *C. ficifolia*’s 2524 SNPs dataset using the *populations* module v1.44 of *Stacks* [59]. We also performed a Principal Component Analysis (PCA, uncentered and unscaled; using library *adegenet* v2.1.3 in R software) [60] and an assignment analysis with *Admixture* v1.3.0, using a 25-fold cross-validation for *K* = 1 to *K* = 15 [61].

The historical demography was inferred based on the analysis of the folded site frequency spectrum (SFS), using unrelated individuals and with a down-sampled dataset of 18 haploid samples to avoid missing data and maximize the number of segregating sites [62]. We then used the composite-likelihood approach of Excoffier et al. [63] implemented in *FastSimCoal* v2.6 for comparing the three models of increasing complexity: constant population size, one demographic change, and two demographic changes. At each demographic change, the new population size was estimated according to the likelihood of the data, which means that the resulting values could either reveal an increase (if *Ncurr* < *Nanc*) or a decrease in population size backwards in time (if *Ncurr* > *Nanc*). The best model was selected according to the Akaike Information Criterion (AIC) [64] and confirmed by comparing the likelihood distribution of the observed SFS using the best parameters between the first- and second-ranked models. Parameter estimation settings are described in the electronic Appendix A.

The hypothesis of a population expansion was also evaluated with our plastome dataset using the mismatch distribution of pairwise differences among haplotypes [65]. We used *Arlequin* v.3.11 [66] for testing the goodness-of-fit between the observed and the expected mismatch distribution under population growth using the sum of squared differences (SSD) statistic, using only SNPs that fell in non-coding regions (200 SNPs). The *τ* parameter computed from the observed mismatch distribution was used for estimating the time to expansion *t* with the formula *t* = $\frac{\tau }{2(m_{T}*\mu )}$, where *m_T_* is the length of the sequence and *μ* is the mutation rate per nucleotide per year [65]. Here, *m_T_* = 44,143 bp (calculated from the *.loci file of *ipyrad*), and *μ* is the mutation rates of Aguirre-Dugua et al. [57] for *Cucurbita* non-coding plastome regions *trnL-trnF* (*μ* = 0.0061 substitutions/site/million years) and *rpl20-rps12* (*μ* = 0.0032 substitutions/site/million years) as lower and upper bounds. 

*Genealogical relationships and gene flow among taxa*—Genetic groupings in the five-taxa dataset were identified using PCA and *Admixture* as previously described.

The occurrence of gene flow between *C. ficifolia* and its wild relatives was assessed using two methods. The first method was the ABBA-BABA test based on Paterson’s *D* statistic [67] using the *Dsuite* v0.4 software [68]. This test evaluates the null hypothesis of no gene flow and assesses its significance with a standard black-jackknife procedure. Based on the genetic groupings identified with the PCA and *Admixture*, *C. ficifolia*, *C. radicans*, and *C. pedatifolia* were treated as separate groups, whereas *C. foetidissima* and *C.* x *scabridifolia* were pooled in a fourth group (hereafter named *foetscabri*). Three trios were tested (following the format ((P1, P2), P3): ((*pedatifolia*, *foetscabri*), *ficifolia*), ((*radicans*, *foetscabri*), *ficifolia*), ((*radicans*, *pedatifolia*), *ficifolia*).

The second method was the coalescent approach of Excoffier et al. [63] based on the unfolded SFS implemented in *FastSimCoal* v2.6. Considering the still obscure phylogenetic position of *C.* x *scabridifolia*, we first assessed four possible models of relationships between this taxon and its closest relatives: simultaneous divergence of the three taxa (model I), *C.* x *scabridifolia* as sister to *C. foetidissima* (model II), *C.* x *scabridifolia* as sister to *C. pedatifolia* (model III), and *C.* x *scabridifolia* as a hybrid of *C. pedatifolia* and *C. foetidissima* (model IV). Each model was run twice: in the absence of gene flow and considering gene flow from each putative parental species to *C.* x *scabridifolia*. Once the best model was chosen for these three taxa, the resulting topology was included in a second stage of model selection that followed the topology (*C. foetidissima*, *C.* x *scabridifolia*, *C. pedatifolia*), *C. radicans*), *C. ficifolia*), comparing models with and without gene flow between *C. ficifolia* and each of its wild relatives (parameter estimation settings and model topologies are shown in electronic Appendix A).

Regarding plastome data, individual sequences were used to build a Maximum-Likelihood (ML) phylogenetic tree with *PhyML* v3.0 [69] with a GTR+G substitution model with 4 substitution rate classes, a Gamma shape parameter estimated from the data and 1000 bootstrap steps to assess branch support.

## 3. Results

### 3.1. Cucurbita ficifolia Genome

The *C. ficifolia* genome combined Illumina HiSeq4000 (90x coverage) and PacBio Sequel (97x coverage) reads for a final assembly in 640 contigs with an N50 contig size of 2.67 Mbp and an L50 of 27 contigs (Appendix A). We were able to anchor 97.4% of the assembly into 20 scaffolds corresponding to each of the chromosomes. We also detected 93% of complete BUSCOs, 1.6% of fragmented BUSCOs, and 5.4% of missing BUSCOs, which indicate a quality of the assembly comparable to other published *Cucurbita* genomes [40,45,50].

### 3.2. Cucurbita ficifolia Diversity and Demographics

We detected low differentiation among states (*F_ST_* = 0.074), and high levels of nucleotide diversity (π = 0.225). *Cucurbita ficifolia* samples showed lower observed than expected heterozygosity, leading to a high inbreeding coefficient (*F_IS_* = 0.233; Table 2). The *Admixture* analysis supported a single gene cluster at *K* = 1, with some differentiation of samples originating in Chiapas, Oaxaca, and Tlaxcala at *K* = 2, as well as Estado de México at *K* = 3 (electronic Appendix A). In the PCA plot, these samples were the ones on the rightmost side of PC1 (Figure 3a). Chiapas and Oaxaca correspond to the southernmost limit of the crop’s distribution in Mexico, whereas Tlaxcala and Estado de México are in Central Mexico (Figure 2b), but these samples did not display any grouping on the plastome ML phylogeny (Figure 3b).

The AIC values of the demographic scenarios of Mexican *C. ficifolia* support the occurrence of two demographic changes: a population expansion followed by a bottleneck (towards the present, model no. 3 in Figure 3c; Table 3). The likelihood of the SFS under model no. 3 was also consistently higher and did not overlap with the likelihood distribution of the SFS under the one-demographic change model ranked no. 2 (electronic Appendix A).

The estimated parameter values of the selected model no. 3 suggest that the *C. ficifolia* population grew from an ancestral population size of *Nanc2* = 215,232 (95% CI: 1787, 92,507) to a population size that was around two times larger (*Nanc1* = 359,555; 95% CI: 58,822, 310,978) about 123,660 generations ago (CI: 15,429, 125,757) (Figure 3d). A second demographic change occurred about 2409 generations ago (95% CI: 464, 12,393) when a population contraction occurred that reduced the nuclear effective population size to *Ncurr* = 43,229 (95% CI: 7559, 62,269) (Figure 3d). Considering the confidence intervals, the first demographic change (from past to present) represented a 2.4- to 9.2-fold growth of *Nanc2* to *Nanc1*, whereas the second demographic change encompassed a population reduction of 0.014 to 0.641 times from *Nanc1* to *Ncurr* (proportions directly calculated from parameter values shown in Figure 3d).

According to plastome data, *Cucurbita ficifolia* samples were grouped in a single, strongly supported clade, where geographical patterns were absent, and samples from the same state were found in different branches (Figure 3a). Moreover, additional Sanger sequences of three non-coding cpDNA regions, where no variation was found, confirmed the nearly homogeneous composition of maternal lineages (electronic Appendix A).

The observed mismatch distribution of pairwise differences was not significantly different from the expected distribution under the null model of a population expansion (SSD = 0.00048, *p* = 0.922; electronic Appendix A). The *τ* parameter had a value of *τ* = 1.027 (95% CI: 0.0, 2.789), which was translated to a time to demographic expansion of *t* = 1.9069 × 10^−3^ to 3.6352 × 10^−3^ million years ago, i.e., a time frame of 1906 to 3635 years ago. When considering the 95% CI of the *τ* parameter, the estimated time frame to expansion can be extended as far as 5178 to 9872 years ago (when *τ* = 2.789).

### 3.3. Among-Taxa Differentiation and Gene Flow

The principal component analysis showed that *C. ficifolia* is the most differentiated taxon among those included in our study, clearly separated from its wild xerophytic relatives in the first principal component (PC1, Figure 4a). Then, the second principal component (PC2) separated *C. radicans* from *C. pedatifolia*, *C. foetidissima*, and *C.* x *scabridifolia* (Figure 4b).

An additional PCA computed on the same SNP matrix but excluding *C. ficifolia* and *C. radicans* reveals a closer relationship of *C.* x *scabridifolia* to *C. foetidissima* than to *C. pedatifolia* (Figure 4c), a relationship that is also observed in the gene pools identified by *Admixture* at *K* = 3, *K* = 4 and *K* = 5 (electronic Appendix A).

Model selection on the four possible evolutionary relationships among taxa belonging to the *foetidissima* group favoured the scenario where *C. foetidissima*, *C. pedatifolia,* and the putative hybrid *C.* x *scabridifolia* diverge simultaneously from a common ancestor in the presence of gene flow from both *C. foetidissima* and *C. pedatifolia* to *C.* x *scabridifolia* (Model I). The second-best supported scenario was the one where *C.* x *scabridifolia* is sister to *C. foetidissima* but receives gene flow from *C. pedatifolia* (Model II; Appendix A, Appendix A).

In our second stage of model selection for assessing the occurrence of gene flow among the domesticated taxon and its wild relatives (Figure 5), the AIC values of the models with gene flow were larger than the AIC values of the models without gene flow. The best-supported model was the one where there is gene flow between *C. ficifolia* and *C. radicans* (Table 4).

In contrast, the ABBA-BABA test did not reject the null hypothesis of no gene flow, thus suggesting no allele introgression among wild xerophytic taxa and *C. ficifolia* (Table 5).

On the other hand, phylogenetic relationships among taxa based on plastome data showed clearly delimited monophyletic species with strong bootstrap support, in agreement with published data. The *foetidissima* group was recovered as a well-supported clade, divided into two lineages: *C. foetidissima* and (*C. pedatifolia, C.* x *scabridifolia*). Plastome data, therefore, supports *C.* x *scabridifolia* samples as belonging to the *C. pedatifolia* lineage (maternally, for the plastome is inherited via seed) and sister to southern *C. pedatifolia* samples from Puebla and Oaxaca (Appendix A). However, *C.* x *scabridifolia* samples are also shown as a diverse, non-monophyletic group.

## 4. Discussion

### 4.1. Diversity and History of C. ficifolia in Mexico

We estimated that the cultivated populations of *C. ficifolia* in Mexico experienced a reduction in effective population size *N_e_* ca. 2409 generations ago, which is correlated with high inbreeding levels (*F_IS_* = 0.233) and the assignment of all Mexican individuals to a single gene cluster (*K* = 1). However, we also detected high levels of nucleotide diversity (π = 0.225), and similar levels of observed heterozygosity (*H_O_* = 0.148) compared to other native squashes such as *C. argyrosperma* subsp. *argyrosperma* and *C. pepo* subsp. *pepo* (π = 0.197 and 0.095; *H_O_* = 0.169 and 0.094, respectively; Table 2). Higher levels of inbreeding seem to occur in several squash crops, for *C. argyrosperma* subsp. *argyrosperma* and *C. pepo* subsp. *pepo* are also more inbred, despite not having experienced a reduction in their overall genetic diversity compared to their wild ancestral taxa (*F_IS_* = 0.034 and 0.116, respectively [23,70]. *Cucurbita moschata*, whose ancestor is unknown, also displays high inbreeding levels (*F_IS_* = 0.18 [71]). This pattern of increased inbreeding accompanied by the preservation of high genetic diversity may be related to its production in home gardens and small-scale traditional agricultural plots (i.e., not being grown in large-scale monocultures) and to *Cucurbita*’s reproductive system that leads to obligate cross-pollination, for the genus produces short-lived monoecious flowers pollinated by specialized *Xenoglossa* and *Peponapis* solitary bees [72]. Additionally, highland habitats are characterized by a less seasonal climate, where constant humidity allows *C. ficifolia* individuals to persist for more than one year without human management, even in what can be considered a feral state, thereby contributing to the long-term resilience of populations.

The reduction in the *N_e_* of Mexican *chilacayote* agrees with its low morphological variability and previous findings with isozymes and RAPDs [32,33]. Interestingly, the contraction of the nuclear effective population size seems to have been accompanied by an expansion of maternal lineages, whose time estimates correspond to the same historical period (ca. 1906–3635 years ago). The expansion based on a small founder population (in contrast with the reduction in a once larger distribution) is further supported by the lack of geographical structure of maternal lineages (Figure 3a and Appendix A), and the moderate levels of population differentiation (*F_ST_* = 0.340).

Our estimates fall within the timeframe of plant domestication in Mesoamerica [36] and are concurrent with *C. ficifolia*’s first archaeological records (5900–5740 calibrated years ago (cal BP) in South America, ca. 1250 BP in Mexico [36,38]; Figure 1). Therefore, these results support the notion that *C. ficifolia* was domesticated later (in Mexico) compared to native *C. pepo* subsp. *pepo* and *C. argyrosperma* subsp. *argyrosperma* (oldest archaeobotanical remains in Mexico dated at 10,000 cal BP and 8700 cal BP, respectively [73,74]).

However, our time estimates based on molecular data should be interpreted cautiously because *C. ficifolia* plants are not strict annuals such as other *Cucurbita* domesticates, making it difficult to translate generation time to calendar years (with a longer life cycle, our estimates would be underestimating the true age of the bottleneck), and our model recurs to instantaneous population changes (Figure 3c) that may not adequately capture the nature of domestication as a gradual process [75].

Additionally, as long as a wild progenitor is unknown and genetic data are not available for *C. ficifolia*’s South American populations, it is difficult to assess if Mexican *C. ficifolia*’s contraction of nuclear effective population size (a) reflects a domestication bottleneck that impacted the entire crop, including Mesoamerican and South American populations; (b) is the product of a small founder population associated with the introduction of the crop to Mesoamerica from South America; (c) is the product of genomic rearrangements predating domestication (see below); or (d) has resulted from strong selective pressures associated with its unique cool and moist habitat. Moreover, these hypotheses are not necessarily exclusive.

We hope that future studies focused on the selection and management practices of *C. ficifolia* cultivation in both Mexico and South America, as well as comparative genomic analyses with other *Cucurbita* crops, will shed light on the factors that have dynamically produced the current levels of genetic diversity of *C. ficifolia* across the Americas.

### 4.2. Gene Flow between C. ficifolia and Its Wild Relatives

We found evidence of gene flow during the evolution of the *foetidissima* group, but we obtained limited evidence on the occurrence of gene flow between domesticated *C. ficifolia* and its wild relatives. On one hand, the ABBA-BABA test was non-significant for all the trios tested (Table 3). This result may be influenced by the loss of statistical power of this test in the presence of gene flow between the P1 and P2 taxa [67], which is very likely to occur (consider, for instance, that all the models evaluated for the *foetidissima* group have lower AIC values in the absence of gene flow compared to their counterparts, including migration (Appendix A)). A second method based on coalescent model selection (Figure 5) favours the scenario where there is gene flow between *C. ficifolia* and *C. radicans*. Indeed, the *Admixture* results suggest allele sharing between these taxa at *K* = 4 (Appendix A).

According to the phenological patterns described by Lira-Saade [35], *C.* x *scabridifolia*, *C. radicans*, and *C. pedatifolia* flowering co-occurs from June to December, *C. foetidissima* produces flowers all year round, and *C. ficifolia* flowers from August to December. Pollinators that co-occur in the areas of sympatry of *C. ficifolia* and the wild taxa considered here include *Peponapis atrata*, *P. pruinosa*, *P. azteca*, *P. smithii*, and *Xenoglossa fulva* [76,77]. Crop-wild pollen exchange is, therefore, possible on ecological grounds. Furthermore, genetic data support the occurrence of crop-wild gene migration in other taxa of the genus.

Martínez-González et al. [70] showed the occurrence of recent gene flow in *C. pepo*, with a migration rate of *m* = 0.0050 of wild SNP genotypes (*C. pepo* subsp. *fraterna*) migrating into domesticated populations (Mexican *C. pepo* subsp. *pepo* landraces), and of *m* = 0.2502 in the opposite direction. Using nuclear microsatellites, Sánchez-de la Vega et al. [78] estimated an *m* = 0.0068 to 0.1324 between cultivated *C. argyrosperma* subsp. *argyrosperma* and wild *C. argyrosperma* subsp. *sororia*, with a similar proportion of migrants in the opposite direction (i.e., wild to crop, *m* = 0.0067–0.0121), a pattern that was later confirmed using SNP data [23]. In both cases, the wild counterpart is the known ancestor of the crop still cross-compatible with it; the long-term impacts of their spontaneous mating shall be shaped by contrasting selective regimes of natural vs. cultivated environments [79].

Considering the marginal evidence on gene flow obtained in this study, it will be necessary to perform additional analyses to identify the wild alleles introgressed into *C. ficifolia* and assess their impact on the crop from their role in metabolic, defensive, and other potentially adaptive processes. Additionally, our evidence is limited to nuclear alleles. Plastome data show strongly defined maternal lineages per species (Appendix A), which is not surprising considering that the incorporation of wild seeds into cultivated plots is unlikely because *C. ficifolia* is cultivated from seeds obtained from previously cultivated plants.

Altogether, our data support the notion that *C. ficifolia* is a strongly isolated taxon despite its ecogeographic proximity to wild *Cucurbita* taxa, similarity in flowering time, and pollinator availability. According to Castellanos-Morales et al. [28], *C. ficifolia* and the *foetidissima* clade shared a common ancestor that existed about 8.44 million years ago, making this divergence the oldest among all crop-wild relationships in the *Cucurbita* genus. Additionally, postzygotic sterility barriers (i.e., embryo inviability) have been detected through experimental crosses where *C. ficifolia* was unable to produce progeny beyond the F1 generation with other domesticated *Cucurbita*, including *C. moschata*, *C. pepo,* and *C. maxima*, and where crosses with *C. foetidissima* and *C. pedatifolia* produced very low fruit and seed sets, and only when *C. ficifolia* was used as the female parent [32,42].

We hypothesize that such barriers may be related to genomic rearrangements in the *C. ficifolia* lineage. Although chromosome counts in *C. ficifolia* (2*n* = 40) are shared with the remaining taxa of the genus [34], total genomic content is larger (2C value of 0.933 pg) than other *Cucurbita*, including *C. foetidissima* (0.686 pg), *C. pedatifolia* (0.772 pg), *C. pepo* var. *fraterna* (0.865 pg), *C. argyrosperma* var. *argyrosperma* (0.748 pg), and *C. moschata* (0.708 pg [80]. Genomic analyses have also shown that lineage-specific rearrangements have occurred in particular taxa after the whole genome duplication (WGD) at the origin of tribe Cucurbiteae [81]. For instance, *C. moschata* and *C. argyrosperma* display an inversion in chromosome four [23], and *C. moschata* harbours a lower number of homologous gene duplicates compared to other domesticated squashes [81]. Weiling [82] observed that one chromosomal pair of *C. ficifolia* had an affinity for the homologous pair of *C. maxima*, while the other did not, which led him to propose that *C. ficifolia* had a different genome set (AACC) compared to other *Cucurbita* domesticates (AABB). In summary, we hypothesize that the *C. ficifolia* lineage has been subject to chromosomal rearrangements and/or differential gene loss between parental subgenomes that has resulted in strong mispairing during gamete fusion, leading to substantial reproductive incompatibility with other members of the genus. Indeed, copy number variation and differential gene silencing after duplication have been identified as contributing to hybrid sterility and inviability [83].

The first genome of *C. ficifolia* published in this study will allow future comparative studies with other available *Cucurbita* genomes [45,50] that shall shed light on the evolutionary pathways of this diverse and economically relevant genus.

### 4.3. Conservation of the Foetidissima Group

The possibility of crops’ alleles entering wild gene pools in greater proportion than the converse has raised concerns about the potential risk of extinction of wild populations due to the reduction in local fitness, which may lead to a reduction in population growth rates (i.e., demographic swamping) or to the replacement of pure wild genotypes via fertile hybrids (i.e., genetic swamping [84,85,86]). This issue is relevant for wild *Cucurbita* taxa characterized by low population densities and medium-to-small distribution ranges (also expected to be reduced in the face of climate change [87]). Here, the estimated migration rates between cultivated *C. ficifolia* and xerophytic taxa (1.10 × 10^−4^ with *C. radicans* and 6.10 × 10^−6^ with *C. pedatifolia*; Figure 5) are lower than those previously described in other *Cucurbita* crops. Given this result, we consider that wild xerophytic cucurbits are not threatened by crop-to-wild asymmetric gene flow, but that environmental degradation and pollinator loss are still the greatest risks they face [41].

This is particularly important for *C. radicans*, for this wild relative is the one with the highest likelihood of gene exchange with *C. ficifolia*. This species is currently not considered within the secondary nor tertiary gene pool of *C. ficifolia* [77] and no experimental hybridization trials have been developed to test if they are cross-compatible [42]. *C. radicans* is the most sympatric with *C. ficifolia* (Figure 2) and has been enlisted as Endangered by the IUCN because of a decreasing population trend mostly associated with urbanization and agricultural expansion [41]. *Cucurbita pedatifolia* IUCN assessment points to a lack of information regarding population status (i.e., Data Deficient), whereas *C. foetidissima* is considered of Least Concern [88]. All four wild taxa considered in this study are the ones with the greatest conservation priority status of all known wild *Cucurbita* based on in situ and ex situ conservation indicators [41]. They are the focus of breeding efforts to increase resistance to virus and fungi-related diseases in *Cucurbita* cultivars, and to exploit their adaptations to xeric growing conditions [41].

### 4.4. On the Nature of C. x scabridifolia

Our nuclear dataset is the first to provide evidence of the nature of *C.* x *scabridifolia*. Against the hybrid hypothesis, *C.* x *scabridifolia* samples were not found in an intermediate position between the putative parental species in the PCA but formed a recognizable group, and the coalescent model where they are hybrids was ranked third (Appendix A). Maternally inherited plastome data of *C.* x *scabridifolia* showed a strongly supported sister relationship with *C. pedatifolia* (Appendix A), whereas nuclear alleles suggest a closer relationship to *C. foetidissima*, as shown by the *Admixture* analysis (Appendix A) and by the coalescent model where these taxa are sisters (model ranked no. 2, Appendix A).

The genetic affinity between *C.* x *scabridifolia* and *C. foetidissima* is likely the product of ancestral polymorphism retention (and not gene flow) because the model where *C.* x *scabridifolia* is sister to *C. pedatifolia* receiving nuclear alleles via gene flow from *C. foetidissima* ranked fourth (Appendix A). The ancestral polymorphism retention in *C.* x *scabridifolia* and the genetic compatibility among members of the *foetidissima* group [39] may be explained by the recent origin of this clade, at ca. 1.34 million years ago (mya) [28].

The great morphological variability of *C.* x *scabridifolia* (Appendix A) and our genetic evidence suggest the absence of evolutionary cohesiveness within this taxon (already pointed out by Andres [39]). Moreover, we have chosen the best model among those defined a priori, and we do not exclude that the taxa may have undergone other evolutionary pathways not considered here. Future studies at an ecological scale may provide additional evidence on the extent of pollen-mediated gene flow among these closely related xerophytic taxa.

## Figures and Tables

**Figure 1 plants-12-03989-f001:**
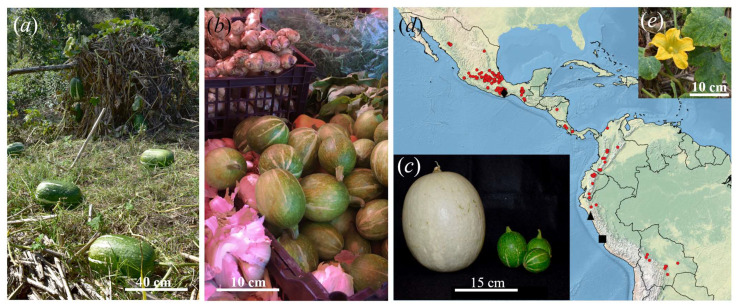
Main characteristics of *Cucurbita ficifolia*. (**a**) Mature fruits in a traditional maize field in northern Puebla; (**b**) immature fruits in a market from Mexico City; (**c**) mature fruit of a white variety (left) and immature fruits (right) sold in Estado de México; (**d**) general distribution of the taxon shown with red dots from Sistema Nacional de Información sobre Biodiversidad (SNIB Conabio) and Global Biodiversity Information (GBIF) (https://doi.org/10.15468/dl.7zd9f8, accessed on 10 July 2022), symbols represent archaeobotanical records (■ La Paloma, 5900–5740 cal BP [36]; ▲ Huaca Prieta ca. 4950 BP [37]; ♦ Guilá Naquitz ca. 1250 BP [38]); (**e**) male flower and leaves.

**Figure 2 plants-12-03989-f002:**
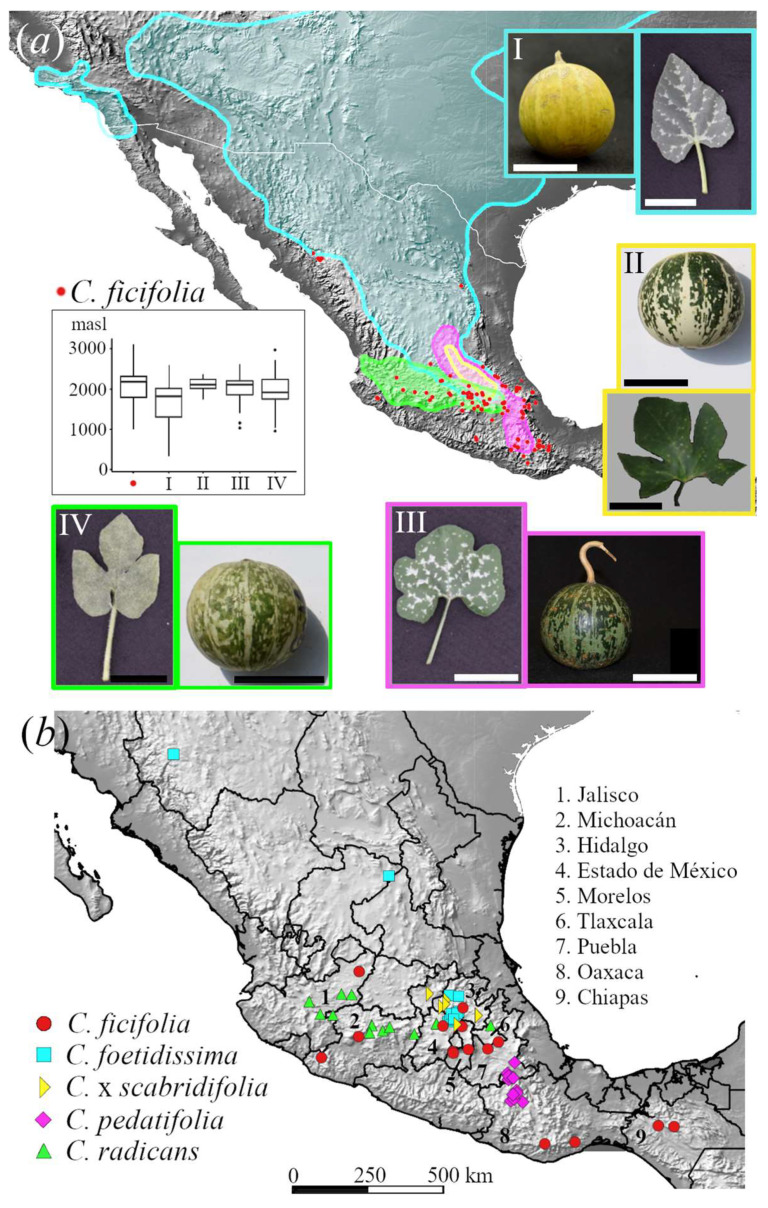
(**a**) Distribution of wild xerophytic *Cucurbita*: (I) *C. foetidissima*, (II) *C.* x *scabridifolia*, (III) *C. pedatifolia*, (IV) *C. radicans*. Records of *C. ficifolia* are shown with red dots. Sources: Sistema Nacional de Información sobre Biodiversidad (SNIB-Conabio) [28,41]. Inset: Elevation (meters above sea level) of records per taxon. Scale bar in I–IV: 5 cm. (**b**) Collection sites of this study with states shown with numbers.

**Figure 3 plants-12-03989-f003:**
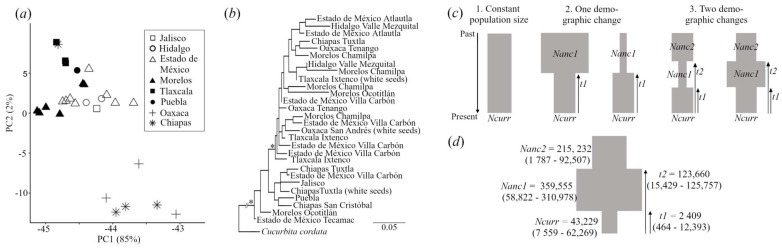
Diversity and historical demography of Mexican *C. ficifolia*. (**a**) Principal component analysis on 2524 unlinked nuclear SNPs. (**b**) ML phylogeny based on 440 plastome variants, branches with bootstrap support >90 are shown with an asterisk. (**c**) Three demographical scenarios tested using nuclear data. *Ncurr* = current effective population size, *Nanc* = ancestral effective population size, and *t* = time (in generations) to population size change. (**d**) Best model with estimated parameter values (95% CI) (Table 3).

**Figure 4 plants-12-03989-f004:**
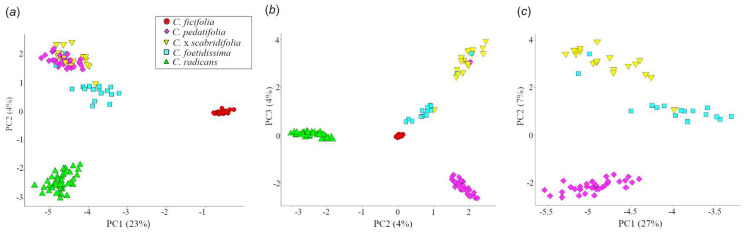
Principal component analysis of the five-taxa dataset with 6292 unlinked nuclear SNPs. (**a**) First and second components. (**b**) Second and third components. (**c**) PCA was performed on *C. foetidissima*, *C. pedatifolia* and *C.* x *scabridifolia* only. Symbols correspond to taxa (see Figure 2).

**Figure 5 plants-12-03989-f005:**
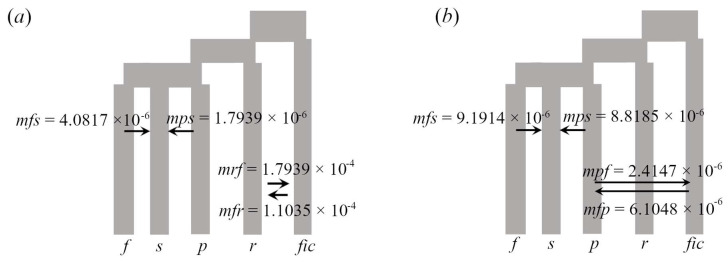
First (**a**) and second (**b**) best-ranked coalescent models considering gene flow between *C. ficifolia* (*fic*) and its relatives *Cucurbita* x *scabridifolia* (*s*), *C. foetidissima* (*f*), *C. pedatifolia* (*p*), and *C. radicans* (*r*). Migration rates are shown with the donor’s and recipient’s first letters; for instance, *mfs* = migration rate from *C. foetidissima* to *C.* x *scabridifolia*.

**Table 1 plants-12-03989-t001:** *Cucurbita* samples included in this study.

Taxon	Status	Habitat	Number of Individuals	Number of Collection Sites
*C. foetidissima*	Wild	Xerophytic	17	12
*C.* x *scabridifolia*	Wild	Xerophytic	19	7
*C. pedatifolia*	Wild	Xerophytic	34	12
*C. radicans*	Wild	Xerophytic	44	14
*C. ficifolia*	Domesticated	Mesophytic	36	15
*C. cordata* (outgroup)	Wild	Xerophytic	11	3

**Table 2 plants-12-03989-t002:** Diversity of Mexican *C. ficifolia* based on 2524 unlinked nuclear SNPs and data from other native domesticated squashes. π: nucleotide diversity; *H_E_*: expected heterozygosity; *H_O_*: observed heterozygosity; *F_IS_*: inbreeding coefficient; var: variance.

Taxon	*π* (var)	*H_E_* (var)	*H_O_* (var)	*F_IS_* (var)	Reference
*Cucurbita ficifolia*	0.225 (0.025)	0.226 (0.024)	0.148 (0.016)	0.233 (0.054)	This study
*C. pepo* subsp. *pepo*	0.197 (0.026)	0.196 (0.025)	0.169 (0.023)	0.116 (0.097)	[70]
*C. argyrosperma* subsp. *argyrosperma*	0.095 (0.010)	0.094 (0.012)	0.094 (0.012)	0.034 (0.030)	[23]

**Table 3 plants-12-03989-t003:** Model selection of three demographic scenarios of *C. ficifolia* from Mexico. LnMaxEstLhood = Maximum estimated likelihood of the SFS. Nparams = number of estimated parameters. AIC = Akaiké Information Criterion. ΔAIC = Difference between the lowest AIC and the AIC of each model.

Model	Historical Demography	Ln MaxEstLhood	Nparams	AIC	ΔAIC	Rank
1	Constant population size	−5144.763	1	10,291.53	−3.48	3
2	One demographic change	−5142.073	3	10,290.15	−2.1	2
3	Two demographic changes	−5139.025	5	10,288.05	0	1

**Table 4 plants-12-03989-t004:** Model selection of scenarios with and without gene flow between *C. ficifolia* and xerophytic wild relatives in Mexico (Figure 5). LnMaxEstLhood = Maximum likelihood of the model; Nparams = number of estimated parameters; AIC: Akaike Information Criterion; ΔAIC = Difference between the AIC of the model and the AIC of the best model among those evaluated. Bold text indicates the two best models.

Model	Gene Flow between *C. ficifolia* and Wild Relatives	Ln MaxEstLhood	Nparams	AIC	ΔAIC	Rank
I.1	Absent	−53,713.31	13	107,452.6	4883.5	5
I.2	*C. ficifolia*- *C. foetidissima*	−52,057.21	15	104,144.4	1575.3	3
I.3	*C. ficifolia*- *C.* x *scabridifolia*	−52,272.53	15	104,575.1	2006.0	4
**I.4**	***C. ficifolia***- ***C. pedatifolia***	**−51,958.6**	**15**	**103,947.2**	**1378.1**	**2**
**I.5**	***C. ficifolia***- ***C. radicans***	**−51,269.55**	**15**	**102,569.1**	**0**	**1**

**Table 5 plants-12-03989-t005:** Statistics of the ABBA-BABA introgression test for *Cucurbita* in Mexico (following the format ((P1, P2), P3), using *C. cordata* as an outgroup.

Trio			BBAA Frequency	ABBA Frequency	BABA Frequency	D-Statistic	Z-Score	*p*-Value
P1	P2	P3						
*pedatifolia*	*foetscabri*	*ficifolia*	4.43434	3.64444	3.55652	0.0122084	0.804987	0.2104 ^ns^
*radicans*	*foetscabri*	*ficifolia*	3.92942	3.61323	3.51278	0.0140957	0.845496	0.1989 ^ns^
*radicans*	*pedatifolia*	*ficifolia*	4.35583	3.87077	3.85665	0.00182715	0.109742	0.4563 ^ns^

ns = non-significant. Bonferroni correction for three simultaneous tests: *p* = 0.05/3 = 0.016.

## Data Availability

The whole genome assembly of *Cucurbita ficifolia* has been deposited at DDBJ/ENA/GenBank under the accession JASJUX010000000. The SRA accessions of the genome and the analysed samples are available at the National Center of Biotechnology Information under the BioProject accession PRJNA485527 (Appendix A). The code for bioinformatic analyses has been included as part of the electronic Appendix A.

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
