# Peer review of "Population Genomics of Domesticated Cucurbita ficifolia Reveals a Recent Bottleneck and Low Gene Flow with Wild Relatives"

_plants, 2023, doi:10.3390/plants12233989_

Round 1

Reviewer 2 Report

The study focuses on Cucurbita ficifolia, a type of squash cultivated from Mexico to Bolivia, with limited compatibility with the wild xerophytic Cucurbita found in Mexico’s highlands. The research successfully established the reference genome of C. ficifolia, examined its genetic diversity and historical demography in Mexico, and explored gene flow between C. ficifolia and related species through genetic markers.

The findings suggest a recent spread of C. ficifolia in Mexico, revealing significant genetic diversity as well as inbreeding. The study also indicates restricted gene flow with closely related species, underscoring the isolated lineage of C. ficifolia. As a result, the research offers a comprehensive analysis of Cucurbita ficifolia's genetic diversity and historical demography, thereby enhancing our comprehension of its evolutionary path and domestication process.

The study's objectives are lucid, and the manuscript, while of interest, has no major issues. Nevertheless, the manuscript's intricate sentence structures could potentially impede readers' comprehension. To enhance readability, it is recommended that the complexity of the statements be reduced, thus aiding the overall readability for a wider readership.

The manuscript's intricate sentence structures could potentially impede readers' comprehension. To enhance readability, it is recommended that the complexity of the statements be reduced, thus aiding the overall readability for a wider readership.

Reviewer 3 Report

The manuscript is overall very easy to read but I believe some issues need to be solved:

- The idea that domestication is a solid process to study evolutionary processes that act on populations is very biased. The authors stated that "The evolutionary process of plant domestication has long been considered a case study for understanding the evolutionary forces that guide population differentiation and speciation, i.e., mutation, selection, genetic drift, and gene flow" . However, domestication is the result of a separation of a species from its natural ecological context, and the selective forces that act on domesticated species depend more on artificial selection than on evolutionary forces. Thus, and because this concept extends very widely in the introduction, I would recommend the authors to re-phrase and explain this better.

- Aims are not clear. The authors stated that "we assembled a high-quality reference genome for the figleaf gourd C. ficifolia and generated a robust dataset of single nucleotide polymorphisms (SNPs) that includes both plastome (i.e., chloroplast’s genome) and nuclear variants to evaluate the diversity and demographic history of C. ficifolia and assess the occurrence of historical gene flow between this crop and its closest wild relatives in Mexico".  Please specifically state the wild relatives. 

- I believe C. ficifolia is a diploid species but the authors should make a clear statement about this. 

- It is not clear how many samples were sequenced. And how many replicates?

- The presence of gene flow is not particularly found in results (in the sense that other options might also occur). Also, the existence of the hybrid could be compatible with a polyploid complex? Can the authors explain this better?

- The format of references ramble throughout the paper. Please check this. 
